# A Statistical Approach for the Integration of Multi-Temporal InSAR and GNSS-PPP Ground Deformation Measurements

**DOI:** 10.3390/s24010043

**Published:** 2023-12-20

**Authors:** Ahmet Delen, Fusun Balik Sanli, Saygin Abdikan, Ali Hasan Dogan, Utkan Mustafa Durdag, Taylan Ocalan, Bahattin Erdogan, Fabiana Calò, Antonio Pepe

**Affiliations:** 1Department of Geomatics Engineering, Gaziosmanpasa University, 60150 Tokat, Türkiye; ahmet.delen@gop.edu.tr (A.D.); alihasan.dogan@gop.edu.tr (A.H.D.); 2Department of Geomatic Engineering, Yildiz Technical University, 34220 Esenler, Istanbul, Türkiye; fbalik@yildiz.edu.tr (F.B.S.); tocalan@yildiz.edu.tr (T.O.); berdogan@yildiz.edu.tr (B.E.); 3Department of Geomatics Engineering, Hacettepe University, 06800 Ankara, Türkiye; sayginabdikan@hacettepe.edu.tr; 4Department of Geomatic Engineering, Artvin Coruh University, 08100 Artvin, Türkiye; umdurdag@artvin.edu.tr; 5Institute for the Electromagnetic Sensing of the Environment (IREA), National Research Council (CNR) of Italy, Via Diocleziano 328, 80124 Napoli, Italy; calo.f@irea.cnr.it

**Keywords:** PS InSAR, deformation, ALOS-2, Sentinel-1, GNSS-PPP, statistical analysis

## Abstract

Determining and monitoring ground deformations is critical for hazard management studies, especially in megacities, and these studies might help prevent future disaster conditions and save many lives. In recent years, the Golden Horn, located in the southeast of the European part of Istanbul within a UNESCO-protected region, has experienced significant changes and regional deformations linked to rapid population growth, infrastructure work, and tramway construction. In this study, we used Interferometric Synthetic Aperture Radar (InSAR) and Global Navigation Satellite System (GNSS) techniques to investigate the ground deformations along the Golden Horn coastlines. The investigated periods are between 2015 and 2020 and 2017 and 2020 for InSAR and GNSS, respectively. For the InSAR analyses, we used sequences of multi-temporal synthetic aperture radar (SAR) images collected by the Sentinel-1 and ALOS-2 satellites. The ground displacement products (i.e., time series and velocity maps) were then cross-compared with those achievable using the Precise Point Positioning (PPP) technique for the GNSS solutions, which can provide precise positions with a single receiver. In the proposed analysis, we compared the ground displacement velocities obtained by both methods by computing the standard deviations of the difference between the relevant observations considering a weighted least square estimation procedure. Additionally, we identified five circle buffers with different radii ranging between 50 m and 250 m for selecting the most appropriate coherent points to conduct the cross-comparison analysis. Moreover, a vertical displacement rate map was produced. The comparison of the vertical ground velocities derived from PPP and InSAR demonstrates that the PPP technique is valuable. For the coherent stations, the vertical displacement rates vary between −4.86 mm/yr and −23.58 mm/yr and −9.50 and −27.77 mm/yr for InSAR and GNSS, respectively.

## 1. Introduction

Istanbul is the province with the lowest per capita area in Turkey. With a population of more than 15 million, it is called Turkey’s megacity. This city, which has a large population in a limited area, is vulnerable to natural and artificial threats. The most important of these are due to earthquakes, which are still in memory and cause severe socio-economic damage when they occur. In parallel with the city’s development, local deformations that may increase the severity of possible disasters also have appeared. In the literature, deformations after earthquakes have primarily been examined for Istanbul using Synthetic Aperture Radar (SAR) techniques. Also, landslides and ground deformations that occur over time have been studied. Aslan et al. [1] calculated an InSAR time series using radar data from ERS-1, ERS-2, Envisat, Sentinel-1A, and Sentinel-1B over approximately 25 years. In the study, vertical ground motions at rates changing between 5 ± 1.2 mm/yr to 15 ± 2.1 mm/yr were detected throughout Istanbul. In the study by Imamoglu et al. [2], ground deformations were detected in Istanbul with Sentinel-1 radar data from October 2014 to October 2017. The size of the deformations was found to be approximately 8 mm/yr. Bayik et al. [3] determined the landslides that occurred between 2015 and 2020 in the Istanbul region with ALOS-2 and Sentinel-1 data. In addition, they supported their study with GNSS observations. According to the results, the horizontal and vertical ground displacements were determined to be in the −10 to 6 mm range. In the study by Calò et al. [4], TerraSAR-X data were used between November 2010 and 24 June 2012 to determine the ground deformations in the Istanbul Golden Horn region. As a result of the study, the areas under potential risk due to deformations were determined. Yagmur et al. [5] computed the displacement over the new construction of Istanbul Airport using Sentinel-1 data. Halicioglu et al. [6] studied the deformation of metro line stations in Istanbul with levelling and Sentinel-1 data. Various data, time intervals, and InSAR techniques used in previous studies have revealed the deformation rate and its variation over time for Istanbul. However, there are limited studies in the literature to determine deformations based on C- and L-band data in this region.

Data obtained with SAR satellite sensors can be used to observe city deformations. These data, processed with SAR Interferometry techniques, efficiently monitor local deformations in cities due to their high spatial and temporal resolution. In the literature, comparisons have usually been made by taking the closest coherent point to the GNSS stations or considering the average values of some nearest coherent points. Similarly, a suitable radius size can be defined with the coherent points determined at different radii. A previous study by Bui et al. [7] determined a diameter of 100 m around the GNSS station in the InSAR study carried out between 2016 and 2020. Fabris et al. [8] determined the displacements of the detected Persistent Scatterer (PS) points in clusters of 50, 100, 150, and 200 m radii using Sentinel-1 and COSMO-SkyMed SAR data. In most of the studies in the literature, network solutions have been used to analyze GNSS observations. For example, Wang et al. [9] combined GNSS velocities obtained from network solutions and InSAR rate maps to form velocity and strain rate fields for south-central Tibet. Li et al. [10] used GNSS and InSAR data for the deformation monitoring and analysis of Kunyang phosphate mine fusion. They concluded that the overall subsidence trend of GNSS data obtained from network solutions is consistent with the InSAR results.

Moreover, Liu et al. [11] used InSAR, GNSS, and levelling observations to generate a three-dimensional deformation field for the Ordos Block, China. In recent years, however, the Precise Point Positioning (PPP) technique, which has a higher positioning consistency than the network solutions [12], has also become popular. Besides, PPP provides global reference frame solutions; hence, it does not include local distortions associated with differential positioning techniques [13]. Thus, it has been preferred for monitoring ground deformations and to determine tectonic movements as an alternative to network solutions [14,15,16].

Since the coastal areas of the Golden Horn are filling ground, a subsidence risk might be expected in these regions. Hence, one of our aims in this study is to determine vertical ground deformations using the Persistent Scatterer InSAR (PSI) technique on multi-temporal C- and L-band SAR data. Previous studies conducted in the region have obtained surface movements with high-resolution X-band [4] and C-band [1,2] data. However, only SAR data were used, and GNSS measurements were not exploited. Unlike other studies, L-band satellite data were used in this study, and GNSS measurements were also performed. In addition, the detected PS points are compared around GNSS campaign stations. We analyzed the GNSS observations based on the PPP technique to see the method’s ability. The most suitable radius was determined by looking at the correlation values of the GNSS station with different radii. Additionally, one of the outlier detection methods was performed on the PS points included in the analysis with the aim to remove the inconsistent points. The weight of the points was calculated with the standard deviations of the obtained PS points. The weighted velocity estimation of the PS points was performed. Moreover, we calculated the vertical displacement rate of the stations based on the three InSAR datasets to determine the subsidence and uplift zones. The vertical displacement rates obtained from InSAR and GNSS methods were compared at the final stage.

## 2. Materials and Methods

### 2.1. Study Area

In this study, we focused on determining the ground deformations in the Golden Horn, one of the coastal regions of Istanbul. The city, with a population of over 15 million, plays a crucial role in culture, politics, and the economy. Specifically, the study area is the Golden Horn coastline, a natural harbor with a key position for transportation. Moreover, Golden Horn, known as the trading center of old Istanbul, is one of the most popular tourist areas of the city. The study area is also intertwined with the region under UNESCO protection. It is crucial to note that the coastlines of the Golden Horn do not have a solid geological structure due to the alluvial structure and artificial ground fillings. Hence, the region is vulnerable to slowly developing landslides due to its topography and geological structure and is in the southeast of the European part of Istanbul (Figure 1). Within the transportation network of Istanbul, Golden Horn has critical importance since the region serves as a hub for both sea and railway transportation, highlighting its strategic significance in facilitating urban connectivity.

### 2.2. SAR Data

Sentinel-1A and ALOS-2 SAR data were used in this study (Table 1). ALOS-2, operated by the Japan Space Agency (JAXA), is the latest developed and operating satellite among Japanese earth observation satellites. An active microwave radar (PALSAR-2) using the 1.2 GHz frequency range is mounted aboard this satellite [17]. Sentinel-1 is one of Sentinel’s mission satellites, which provides all-weather, day, and night radar imaging for land and ocean services at the C-band under the European Copernicus program. SAR images of Sentinel 1A were collected through both descending and ascending orbits. We used 28 ALOS-2 PALSAR-2 L band images obtained from ascending trajectories. In addition, we utilized 62 Sentinel-1 SAR data acquired with the TOPSAR Interferometric Wide (IW) Swath Mode in Single Look Complex (SLC) format from ascending orbits and 61 Sentinel-1 IW SLC data from descending orbits. For both ascending and descending datasets, VV polarization bands were processed.

### 2.3. GNSS Observations

The Golden Horn GNSS (GHGNSS) network was designed to determine the displacements along Golden Horn. GNSS measurements were started in August 2017 and repeated until March 2020. In August 2017, 14 stations were established according to the deformation areas given by Calò et al. [4] (Figure 2). However, due to the construction activities in the Golden Horn region, the GH12 station was re-established and named GX12. Moreover, two new stations named GH14 and GH17 were established after the fifth campaign. The session duration of the observations is about 6 h, with an interval of 10 s for all stations. The details of the campaigns are shown in Table 2.

### 2.4. Analysis of the GNSS Data

GNSS observations were processed based on the PPP technique, as stated before. This technique, which can provide position information with a single receiver, was first introduced at the end of the 1990s as an alternative Global Navigation Satellite Systems (GNSS)-based positioning technique [18]. Several error sources and correction models should be considered to obtain precise positions when using PPP. Accurate position information and displacements can be obtained using precise orbit and clock products. Since this technique does not need a network structure and provides global reference frame solutions, local deformations can be determined with a single receiver.

PPP solutions were carried out using Trimble CenterPoint RTX online post-processing service. The service supports the processing of dual-frequency code and carrier phase observations of multi-GNSS systems based on the following equations [19].
(1)$$P_{i,k}^{j}=\rho_{i}^{j}+c\Delta t_{i}-c\Delta t^{j}+T_{İ}^{j}+I_{P,i,k}^{j}+b_{P,i,k}-b_{P,k}^{j}+m_{P,i,k}^{j}+\epsilon_{P,i,k}^{j}$$
(2)$$\phi_{i,k}^{j}=\rho_{i}^{j}+c\Delta t_{i}-c\Delta t^{j}+T_{İ}^{j}+I_{\phi ,i,k}^{j}+b_{\phi ,i,k}-b_{\phi ,k}^{j}+\lambda_{k}N_{i,k}^{j}+m_{\phi ,i,k}^{j}+\epsilon_{\phi ,i,k}^{j}$$
where $P_{i,k}^{j}$ and $\phi_{i,k}^{j}$ are code and carrier phase observations, $\rho_{i}^{j}$ is the geometric range, c is speed of light, $\Delta t_{i}$ and $\Delta t^{j}$ are the receiver and satellite clock errors, respectively. $T_{İ}^{j}$ represents tropospheric delay. $I_{P,i,k}^{j}$ and $I_{\phi ,i,k}^{j}{(=-I}_{P,i,k}^{j})$ are the ionospheric delay, $b_{P,i,k}$ and $b_{\phi ,i,k}$ code and carrier phase biases for the receiver, $b_{P,k}^{j}$ are $b_{\phi ,k}^{j}$ code and carrier phase biases for satellite, $\lambda_{k}$ is carrier wavelength of frequency *k*, $N_{i,k}^{j}$ is integer ambiguity, $m_{P,i,k}^{j}$ and $m_{\phi ,i,k}^{j}$ are code and carrier multipaths, $\epsilon_{P,i,k}^{j}$ and $\epsilon_{\phi ,i,k}^{j}$ are the measurement errors for code and carrier phase measurements, respectively.

In the processing, we used both GPS and GLONASS observations. This service uses precise orbits and clocks produced by Trimble RTX. The update rate of the products is 1 Hz [20]. For static positioning, the convergence time is 15 min [19]. Moreover, linear combinations are used to eliminate ionospheric delays. Tropospheric delays and receiver clock errors are estimated in the processing step. In the service, a coordinate system can be selected by the user. We selected the ITRF2014 datum. The coordinates and their covariance matrix were obtained in a 3D Cartesian coordinate system. We then transformed each campaign’s coordinates and covariance matrices to a topocentric coordinate system with respect to each station’s first epoch. Moreover, a 6 h observation session duration, as indicated by Sezer et al. [20], employing the accuracy function provided by Saracoglu and Sanli [21], can yield millimeter-level accuracy.

### 2.5. Analysis of the SAR Data

Ground deformations in the Golden Horn were found using a multi-temporal InSAR analysis. SAR data processing was performed using SarProz software [22]. Before beginning the image processing, a subset selection of the study area was carried out. Considering the definition of the orbit and atmospheric information, the image dated July 2017 was selected as the primary image. Then, the co-registration process was applied to the primary and secondary images.

At the stage of processing the ALOS-2 data, although the temporal and spatial bases of the images dated 5 October 2014, 14 December 2014, and 22 February 2015 were suitable, they were not included in the PSI analysis process due to low coherence and high level of errors in their trajectories. Provided analyses were carried out with 24 images.

Sentinel-1 images acquired at C-band on different orbits (descending and ascending) were obtained from the multi-temporal InSAR analysis conducted within the scope of the study. A set of coherent PS points were independently obtained after the PSI analysis was carried out using the collected SAR datasets. The ground deformations of these points were determined along the Line of Sight (*LOS*) direction. From the ALOS-2 dataset, 29,401 PS points were obtained. From the Sentinel-1 dataset, 33,501 and 33,552 PS points were obtained for ascending and descending trajectories, respectively. Figure 3 shows the spatial distribution of the PS points obtained for ALOS-2, Sentinel-1 descending, and ascending orbit. For all three datasets, PS points were obtained along with GH. The distribution of the PS points for the region was predominantly clustered in the urban area (road, buildings, port, bridge, etc.). For the whole coastal region of the GH, the mean displacement rates are −1.10 mm/yr, −3.82 mm/yr, and −5.08 mm/yr for ALOS-2, Sentinel-1 ascending, and Sentinel-1 descending, respectively. For the northern part of the GH (middle panel of the figure), the mean rates are −5.47 mm/yr, −11.49 mm/yr, and −14.25 mm/yr, respectively.

## 3. Results and Discussion

### 3.1. Statistical Evaluation of InSAR and GNSS-PPP Results

For the InSAR analysis, the ground displacements were obtained along the *LOS* direction. Calò et al. [4] employed one and a half years of TerraSAR-X data from November 2010 to June 2012, utilizing the Small Baseline Subset (SBAS) method. Their analysis revealed deformation in the coastal areas of the GH, resulting in a displacement of approximately 4–5 cm (an average of ~3 cm/year). Moreover, Imamoglu et al. [2] reported a displacement rate of 8 mm/year based on Sentinel-1 data collected between 2014 and 2017. Additionally, Aslan et al. [1] determined a maximum displacement of 10 mm/year using Sentinel-1 data for the same period. The present study corroborates these findings, indicating displacement values in the same region, particularly within the latter section of the GH. Furthermore, it is observed that the observed movement has persisted in this area after 2017.

From the GNSS measurements, the ground displacements are obtained in the topocentric coordinate system, namely, the east, north, and up directions. To compare the results obtained from InSAR and GNSS, we calculated the *LOS* displacements of the GNSS analysis by using the equation given by Hanssen [23] in Equation (3):(3)$$\;LOS=U_{V}\cos\theta -\sin\theta \left[U_{N}\cos\left(\alpha -3\pi /2\right)+U_{E}\sin\left(\alpha -3\pi /2\right)\right]$$
where θ and α denote the angle of incidence of the radar wave and the direction angle of the satellite, $U_{V}$, $U_{N}$, and $U_{E}$ are the displacements obtained from the PPP solutions using the observations from the GNSS campaigns. To calculate the velocities of the stations for the InSAR and GNSS techniques, Equation (4) was used.
(4)$${\;LOS}_{t}={LOS}_{0}+v\;(t-t_{0})$$

Here, ${LOS}_{t}$ and ${LOS}_{0}$ denote the displacements in the *LOS* direction in epoch t and the reference epoch, respectively, and v is the velocity. In the velocity estimation step, we also applied the Pope test to exclude outliers [24].

Before comparing the velocities, we checked the stations’ velocities obtained by the GNSS analysis to see whether the movements were significant. In this step, considerable displacement was seen in nine of the seventeen stations. The subsequent analysis was performed for the nine stations. To compare the InSAR velocities and velocities obtained from the GNSS observations, we calculated the GHGNSS network stations’ InSAR-based velocities using the PS points’ velocities. Here, we created five different circle areas for each station. The centers of the circles were chosen as the GNSS stations, and the radii were determined as 50, 100, 150, 200, and 250 m. Then, we used the PS points within the determined circles. In Table 3 and Table 4, the numbers of the PS points are shown for each station and each circle. As expected, and in the case of the ALOS-2 data (Table 3), the number of PS points increases when the radius increases for the Sentinel-1 data (Table 4).

As shown in Table 3 and Table 4, there are no PS points in some areas within the first and second-smallest circles, i.e., in the 50 m and 100 m radii. As a result, statistical analyses and estimates were carried out for the other circles (150, 200, and 250 m). Next, we calculated the correlations of the displacements of the PS points within the circles and the GNSS stations’ displacements. The correlations between the ALOS-2 and GNSS data are up to values of 0.72, 0.84, and 0.78, respectively. The correlations between Sentinel-1 descending and GNSS are 0.65, 0.75, and 0.61; between Sentinel-1 ascending and GNSS, they are 0.65, 0.82, and 0.75. As can be seen from the correlation values, the best values were reached at a radius of 200 m for all three datasets. Hence, the statistical evaluations for this study will be on the results obtained in the 200 m radius since the highest correlation values were obtained in the 200 m radius circle.

The correlation values for the GH02, GH13, and IGNA stations were close to zero between the GNSS stations and PS points. Thus, these three stations were not included in the evaluation scope due to the errors caused by the observation periods and the possible problems that may have occurred in the campaign measurements. The sources of these errors will be examined in future research.

Under the assumption that the PS points within the circles reflect the GNSS stations’ movements, we took the average of the PS points’ displacements. We calculated the standard deviations for each station. However, some of the PS points within the circles could be outliers. Hence, we applied the median approach to the PS points’ displacements to exclude outliers before taking averages in each epoch for each GNSS station. With a 50% breakdown point, the median approach is one of the most reliable outlier detection methods [20,25,26]. Additionally, the median and median absolute deviation (MAD) are more efficient in outlier detection than the mean and standard deviation. The approach was applied using Equation (5).
(5)$$mad=\left\{\begin{array}{c} \;\;1.2533\times \frac{1}{n}\times \sum \left|X-median(X)\right|,\;\;\;\;\;\;\;\;median(\left|X-median(X)\right|)=0\\ 1.4826\times median(\left|X-median(X)\right|),\;\;\;\;\;\;\;median(\left|X-median(X)\right|)\neq 0 \end{array}\right.$$
where X denotes the displacements of the PS points. And n is the number of the PS points. In this approach, the $\left|X-median(X)\right|$ values are compared with 3 × mad.

In the next step, the InSAR velocities of the GNSS stations for ALOS-2 and Sentinel-1 were estimated by least square estimation (LSE) based on Equation (4) [27]. In the velocity estimation, we also considered the standard deviations of the displacements for each technique. Thus, we took into account the weights of each PS point.

As stated, six stations and three circles were analyzed using statistical methods. However, we simply reported the data of the 200 m radius circle for explanation purposes (Table 5 and Table 6). In Table 5, the velocities obtained from the InSAR methods are shown. The GNSS velocities, transformed into *LOS* directions, have been presented in Table 6.

As seen from Table 5, the velocities of the Sentinel-1 ascending and descending stations fit very well for most of the stations, and their formal errors are below 1 mm/yr. The correlation of the velocities is “0.96”. Moreover, almost all the formal errors of the velocities from ALOS-2 are sub-mm/yr. The correlations of the velocities from ALOS-2 and Sentinel-1 ascending and from ALOS-2 and Sentinel-1 descending are 0.98 and 0.93, respectively. However, the formal errors of the GNSS velocities are higher than the InSAR’s (see Table 6). This might be related to the number of observations. Meanwhile, in the InSAR techniques, there are numerous observations; seven campaigns were carried out for GNSS observations. Nevertheless, the GNSS velocities can be compared with the InSAR technique since, in almost all stations, the directions of the movements are the same for the GNSS and InSAR methods. Moreover, we calculated the correlation values of the displacements between the InSAR and GNSS techniques for all three datasets for 150 m, 200 m, and 250 m. During the correlation calculation, the displacements of the PS points within the distinct buffer zone radii and the displacements derived from the GNSS analyses, which were transformed into the LOS direction, were utilized. These analyses were performed across various buffer zones (Figure 4). For the 200 m radius circle, the highest correlation between the GNSS and InSAR techniques was found at GX12 for the ALOS-2 and Sentinel-1 datasets.

One should note that since the incidence and heading angles are different for the InSAR datasets, the *LOS* values should be compared separately. Hence, we plotted the time series for each dataset, as in Figure 5. It can be seen from the figure that since the *LOS* values derived from the GNSS perfectly fit its trend, the correlation values are also high. One should note that it is possible to compare the combination/integration of the InSAR analysis with the GNSS. We also integrated the three InSAR datasets in this study. The results of the integration are presented in the next section.

In the next step, we tested the significance of the velocity differences between the GNSS and InSAR techniques to determine whether the differences are statistically different from zero. To carry this out, we calculated the ratios of the velocity differences by dividing their standard deviations (Equation (6)).
(6)$$T=\frac{\left|v_{G}-v_{I}\right|}{\sqrt{s_{v_{G}}^{2}+s_{v_{I}}^{2}}}\;\sim \;t_{f,1-\alpha }$$
where T is the test value, $v_{G}$ and $v_{I}$ are the velocities obtained from GNSS and InSAR, and $s_{v_{G}}^{}$ and $s_{v_{I}}^{}$ are the standard deviations of the velocities, respectively. The test values were compared with the critical value with a 95% confidence level.

According to Table 7, the velocity differences for station GH19 are significant. For other stations, all differences are insignificant. Although it can be seen from Table 5 and Table 6 that the velocity differences are up to 1–1.5 cm for GH06, the differences are negligible since the formal error of the GNSS is high compared to the InSAR. The reader should note that the test values are the ratios of the velocity differences obtained by dividing their standard deviations. The test values are significantly lower because the GNSS velocities have more considerable standard deviations than InSARs. Nevertheless, in the stations (except GH06 and GH19), the trends of the GNSS and InSAR techniques show the same direction and fit. The maximum velocity difference is 3.68 mm/yr.

### 3.2. Multi-Orbit/Multi-Frequency SAR Integration

In this subsection, we finally address the problem of decomposing the *LOS*-projected ground displacement rates into the region’s 3D (up–down, east–west, and north–south) displacement rates. Generally speaking, at least three *LOS*-projected displacement rates recovered from the complementary (ascending/descending) orbits are needed to solve this problem. Our work combined the ALOS-2, Sentinel-1 ascending, and Sentinel-1 descending SAR datasets. Nonetheless, since the satellites fly near-polar orbits, the sensitivity of the *LOS* displacements to the north–south components of the deformation is significantly limited, and only the vertical and east–west displacement rates can accurately be retrieved. Indeed, it is known that the sensitivities of the SAR system to the 3D ground motions are as follows:(7)$$\frac{\partial LOS}{\partial U_{V}}=\cos\theta \;,\;\;\;\;\;\;\frac{\partial LOS}{\partial U_{N}}=-\sin\theta \cos\left(\alpha -3\pi /2\right)\;,\;\;\;\;\;\;\frac{\partial LOS}{\partial U_{E}}=-\sin\theta \sin\left(\alpha -3\pi /2\right)$$
where [U_v_, U_E_, U_N_] are the 3D components of the ground deformation, θ is the local incidence angle, and α is the azimuth angle. Accordingly, the observation equation can be written as follows:(8)$$\left[\begin{array}{c} {LOS}_{i}\\ {LOS}_{j}\\ {LOS}_{k} \end{array}\right]=R\left[\begin{array}{c} U_{V}\\ U_{N}\\ U_{E} \end{array}\right]$$
where i, j, and k represent ALOS-2, Sentinel-1 ascending, and Sentinel-1 descending, respectively, [*LOS*_i_, *LOS*_j_, *LOS*_k_]^T^ is the vector of the relevant *LOS*-projected ground-displacement rates, and R is the transformation matrix:(9)$$R=\left[\begin{array}{ccc} \cos\theta_{i} & -\sin\theta_{i}\cos\left(\alpha_{i}-3\pi /2\right) & -\sin\theta_{i}\sin\left(\alpha_{i}-3\pi /2\right)\\ \cos\theta_{j} & -\sin\theta_{j}\cos\left(\alpha_{j}-3\pi /2\right) & -\sin\theta_{j}\sin\left(\alpha_{j}-3\pi /2\right)\\ \cos\theta_{k} & -\sin\theta_{k}\cos\left(\alpha_{k}-3\pi /2\right) & -\sin\theta_{k}\sin\left(\alpha_{k}-3\pi /2\right) \end{array}\right]$$

Instead of applying more sophisticated, time-consuming approaches for the calculation of the 3D ground displacement rates, relying on the combination of the *LOS*-projected ground displacement time series (e.g., see [28,29,30]), in this work, we followed the straightforward strategy outlined in [31], which assumes the deformations are almost linear over time (and this assumption is reliable in our case) and only requires *LOS*-projected ground displacement rates to be available for the three datasets over a common spatial grid. To this aim, the obtained ground displacement rates of the detected PS points were calculated over 50 m × 50 m grid points after masking the incoherent areas. Then, Equation (8) was applied to recover the 3D ground deformation rates.

Figure 6 shows the region’s vertical (up–down) displacement rate map. As can be seen from the figure, the subsidence was obtained mainly in the coastal areas and the northern side of the GH. Moreover, Figure 7 shows a map of the region’s horizontal (east–west) displacement rates.

In the final step, we calculated the vertical displacement rates of the GNSS stations based on the PS points within the 200 m radius circle (Table 8). According to Table 8, it can be concluded that the vertical movements of the stations show the same direction for almost all stations. The velocities obtained from the PPP solutions are consistent with those based on the PS points. The correlation of the velocities is 0.96 (except GH06 and GH19). For GH06 and GH19, we found relatively higher displacement rates with PPP. This might be related to the PS point number within the circle and the displacements of the PS points.

The InSAR and GNSS techniques are frequently used in the determination of surface deformations. Validation between the GNSS and InSAR data outcomes cannot be made with any degree of certainty. While determining the deformations in the Istanbul Golden Horn Region for our study, we used GNSS and InSAR data. The PS points that fell into it were identified by creating a circle with the optimum radius around the GNSS station to compare these two methods. The correlation values of the displacements were computed to determine the radius of this circle enclosing the GNSS station. ALOS-2 and Sentinel-1 datasets were found to have the highest correlation at points with a radius of 200 m. Fabris et al. [8] also investigated the integration of GNSS and InSAR to monitor the land subsidence in the Po River Delta by determining the circle around the GNSS stations. They also selected a 200 m radius circle but without any statistical analysis. In this study, we considered the correlation values to choose the most suitable radius. Moreover, only velocities in the *LOS* direction were analyzed in that study.

## 4. Conclusions

We used the GNSS and InSAR techniques to determine the vertical deformations in the Istanbul Golden Horn region. When previous research was scrutinized, it became clear that the GNNS and InSAR data were validated using one or more PS points close to the GNSS station. Additionally, it was noted that assessments were conducted using PS points inside a circle defined by the GNSS observation station. The optimum radius for the circle formed around the GNSS stations cannot be determined with certainty. This research found the PS points within five different radii to have the highest correlation with the GNSS station in the 200 m radius. A novel aspect of the study was that the PPP technique was used to process the data from the GNSS campaigns, and we identified the outliers from the PS points that would be utilized to calculate the deformations; to these, we applied the median test. Also, in the velocity estimation step, we considered the weights of the displacements by their standard deviations. According to the results, the 200 m radius circle is suitable for the PS points. In the final step, we obtained the vertical displacement rate map for the region and the stations based on the *LOS* values of the InSAR datasets. According to the vertical velocities, the subsidence was seen, especially in the coastal area of the GH. Given the vertical velocities of the PPP solutions and InSAR fit, it can be concluded that the PPP technique can be used for the InSAR and GNSS deformation monitoring studies.

## Figures and Tables

**Figure 1 sensors-24-00043-f001:**
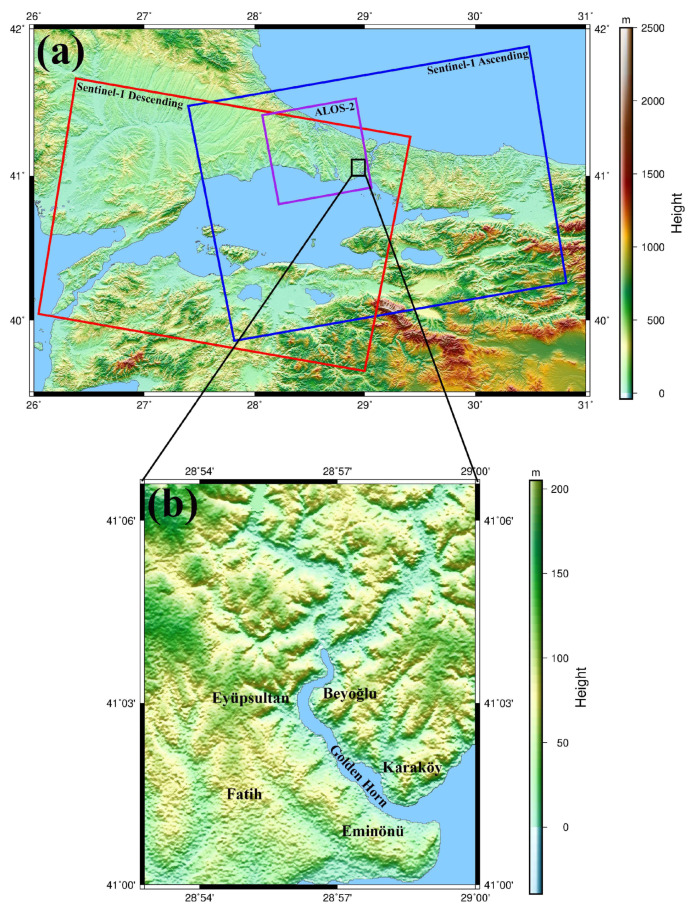
Topography and geography of the studied area. Red, blue, and purple boxes highlight the swaths of the used Sentinel-1 (descending), Sentinel-1 (ascending), and ALOS-2 SAR datasets. A zoomed in view of the height profile of the zone identified by the black rectangle in (**a**) is shown in (**b**).

**Figure 2 sensors-24-00043-f002:**
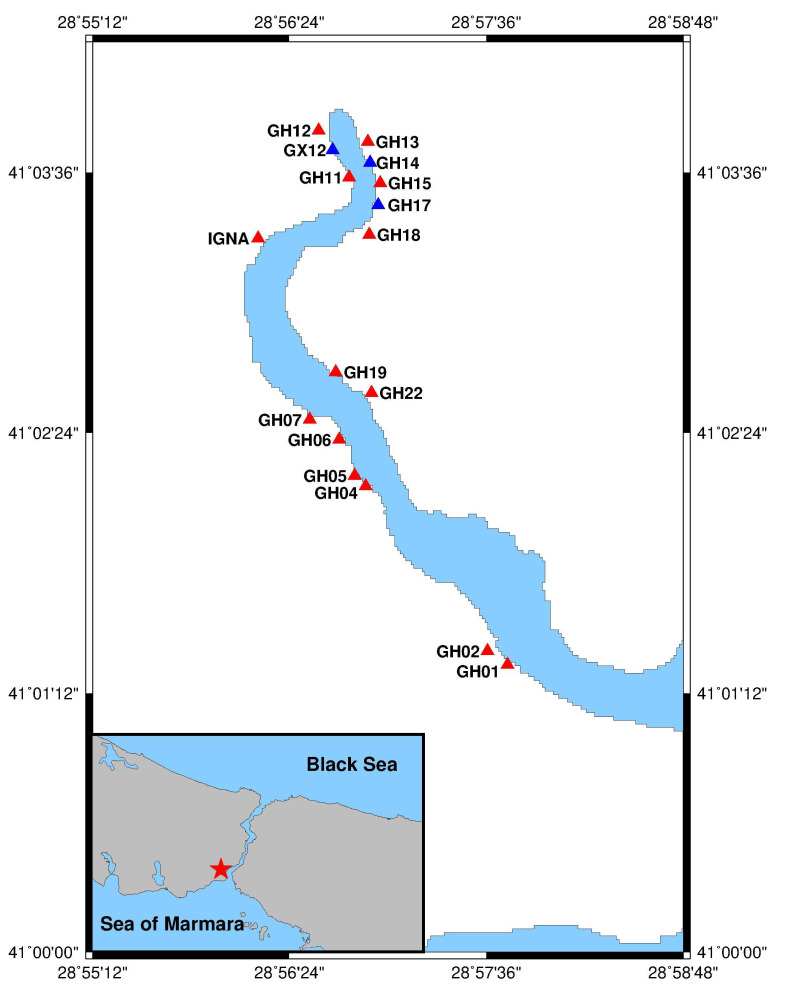
GHGNSS network. Fourteen established stations in August 2017 are colored red, and the others are colored blue. The red star represents the study area.

**Figure 3 sensors-24-00043-f003:**
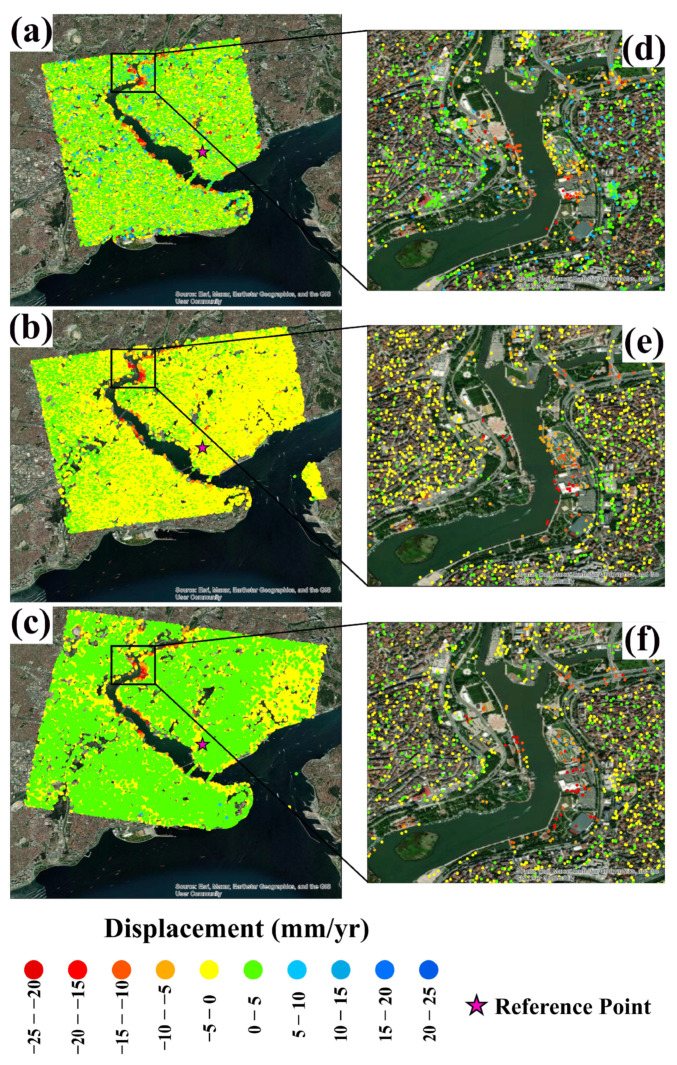
(**a**–**c**) Maps of the ground deformation velocities as obtained for ALOS-2, Sentinel-1 (ascending), and Sentinel-1 (descending) SAR datasets, respectively. The insets in (**d**–**f**) show the distribution of the detected PS points highlighted by the black rectangle in (**a**–**c**).

**Figure 4 sensors-24-00043-f004:**
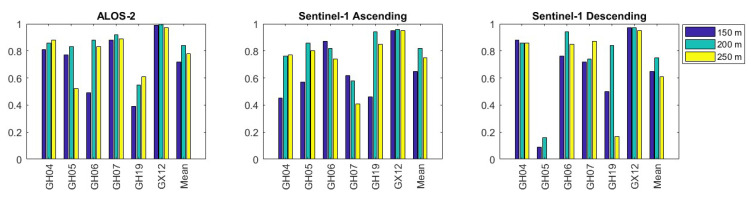
The graph of correlation values. Blue, green and yellow histograms refer to spatial boxes on the ground with a radius of 150 m, 200 m, and 250 m, respectively.

**Figure 5 sensors-24-00043-f005:**
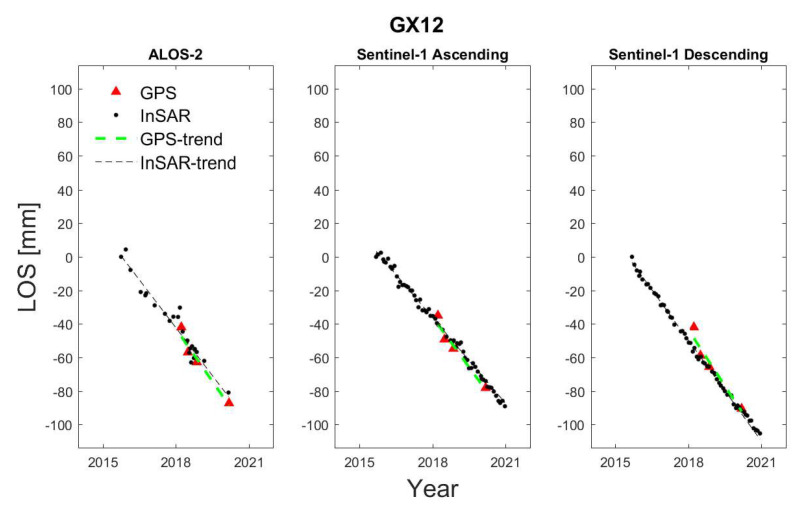
The time series of GX12.

**Figure 6 sensors-24-00043-f006:**
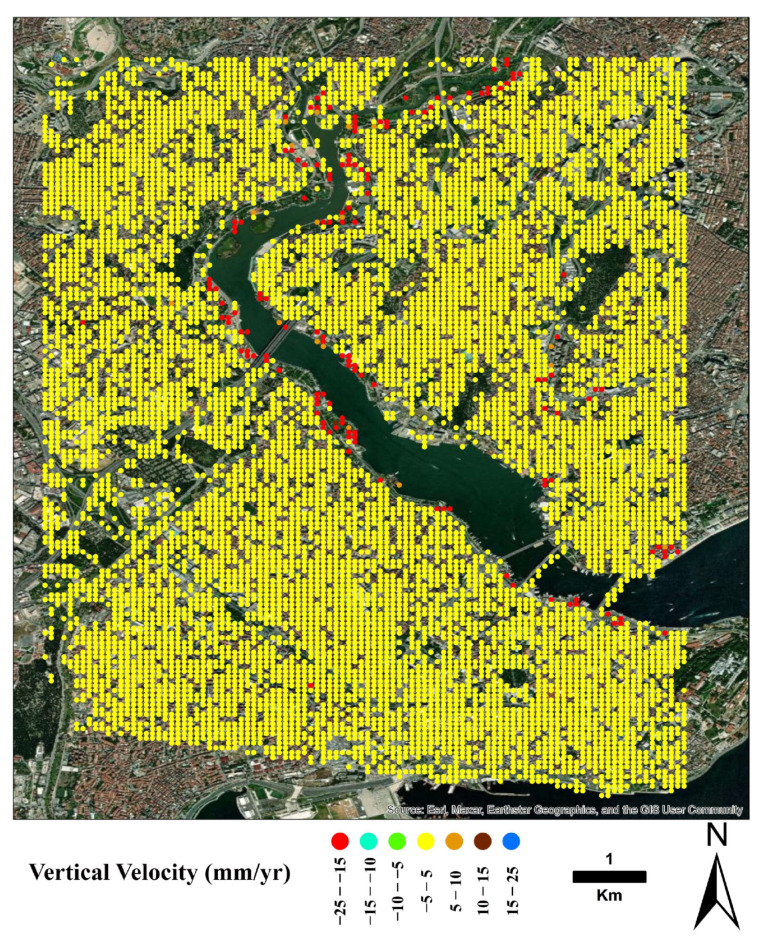
Map of the vertical displacement rate computed over a 50 m × 50 m grid after masking the incoherent areas, interpolating PSI-derived *LOS* ground deformation velocity values and applying the adopted, simple combination method in [31].

**Figure 7 sensors-24-00043-f007:**
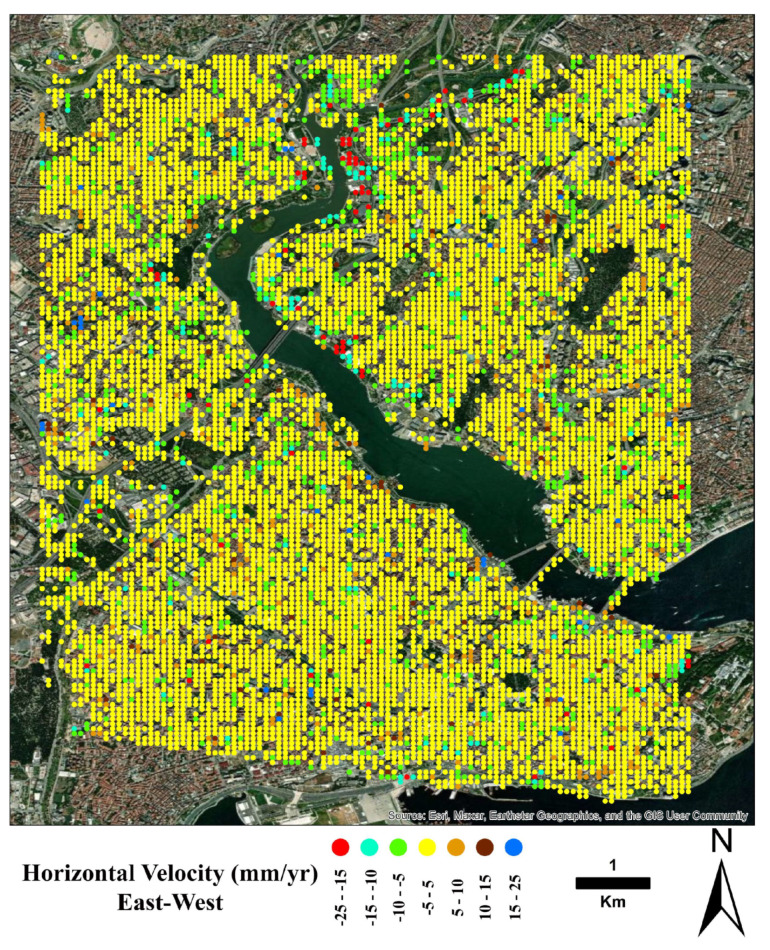
Same as Figure 6 for the east–west ground displacement rates of the region.

**Table 1 sensors-24-00043-t001:** The SAR data used in the study.

	Sentinel-1A		ALOS-2
Orbit pass	Ascending	Descending	Ascending
Wavelength	5.5 cm		24 cm
Acquisition mode	IW		SM3
Incidence angle (deg)	39.3274	38.9379	36.2940
Polarimetry	VV		HH
Period	September 2015–December 2020		October 2015–March 2020
Primary date	2 June 2018	2 April 2018	23 July 2017
Number of images	62	61	28

**Table 2 sensors-24-00043-t002:** The details of the campaigns.

Campaign	Date	Day of Year (DOY)	Measured Sites
1	August 2017	226–227	GH01, GH02, GH04, GH05, GH06, GH07, GH11, GH12, GH13, GH15, GH18, GH19, GH22, IGNA
2	November 2017	322–323	GH01, GH02, GH04, GH05, GH06, GH07, GH11, GH12, GH13, GH15, GH18, GH19, GH22, IGNA
3	March 2018	83–84	GH01, GH02, GH04, GH05, GH06, GH07, GH11, GH13, GH15, GH18, GH19, GH22, GX12, IGNA
4	July 2018	181–182	GH01, GH02, GH04, GH05, GH06, GH07, GH11, GH13, GH15, GH18, GH19, GH22, GX12, IGNA
5	November 2018	314–315	GH01, GH02, GH04, GH05, GH06, GH07, GH13, GH15, GH18, GH19, GH22, GX12, IGNA
6	November 2019	327–328	GH01, GH02, GH05, GH06, GH07, GH13, GH14, GH15, GH17, GH22, GX12, IGNA
7	March 2020	67–68	GH01, GH02, GH04, GH05, GH06, GH07, GH13, GH14, GH15, GH17, GH22, GX12, IGNA

**Table 3 sensors-24-00043-t003:** Number of PS points for the ALOS-2 dataset.

ALOS-2					
Station	50 m	100 m	150 m	200 m	250 m
GH02	1	6	15	26	39
GH04	0	3	13	26	37
GH05	1	5	14	32	52
GH06	0	0	3	8	21
GH07	0	3	8	13	29
GH13	2	14	44	66	100
GH19	2	11	27	48	70
GX12	0	1	16	44	78
IGNA	0	3	9	19	28
*Average*	**0.7**	**5.1**	**16.6**	**31.3**	**50.4**

**Table 4 sensors-24-00043-t004:** Number of PS points for Sentinel-1 datasets.

Station	Sentinel-1 Ascending					Sentinel-1 Descending				
	50 m	100 m	150 m	200 m	250 m	50 m	100 m	150 m	200 m	250 m
GH02	0	1	6	16	33	0	6	15	27	46
GH04	0	1	4	6	16	5	8	9	20	31
GH05	1	2	11	21	42	0	2	17	40	57
GH06	0	2	5	10	24	0	0	6	18	32
GH07	0	5	7	9	17	1	4	7	13	22
GH13	1	3	17	37	63	3	8	10	25	55
GH19	1	3	13	36	58	0	3	12	43	70
GX12	0	0	5	12	38	1	2	7	17	31
IGNA	0	4	7	11	23	3	4	10	14	23
*Average*	**0.3**	**2.3**	**8.3**	**17.6**	**34.9**	**1.4**	**4.1**	**10.3**	**24.1**	**40.8**

**Table 5 sensors-24-00043-t005:** Estimated velocities and formal errors of the PS points within the 200 m buffer zone of the GNSS stations using different InSAR datasets obtained from InSAR analysis (mm/yr).

Station	$v_{ALOS-2}$	$s_{v_{ALOS-2}}^{}$	$v_{Sentinel-1\;Asc.}$	$s_{v_{Sentinel-1\;Asc.}}^{}$	$v_{Sentinel-1\;Desc.}$	$s_{v_{Sentinel-1\;Desc.}}^{}$
GH04	−11.84	0.28	−11.22	0.33	−9.12	0.12
GH05	−8.07	0.27	−10.00	2.02	−9.97	1.04
GH06	−5.13	1.34	−4.70	0.23	−1.41	0.09
GH07	−6.46	0.24	−7.42	0.16	−3.94	0.18
GH19	1.57	0.25	−0.70	0.09	−0.80	0.06
GX12	−19.06	1.02	−17.07	0.28	−20.10	0.22

$v_{i}$: velocity, $s_{i}$: standard deviation of velocity, i: dataset.

**Table 6 sensors-24-00043-t006:** Estimated velocities and formal errors of the GNSS stations, transformed into *LOS* direction for different orbit trajectories, obtained from GNSS analysis (mm/yr).

Station	$v_{ALOS-2}$	$s_{v_{ALOS-2}}^{}$	$v_{Sentinel-1\;Asc.}$	$s_{v_{Sentinel-1\;Asc.}}^{}$	$v_{Sentinel-1\;Desc.}$	$s_{v_{Sentinel-1\;Desc.}}^{}$
GH04	−8.48	3.38	−8.02	3.24	−9.63	3.17
GH05	−10.85	3.65	−10.33	3.50	−11.98	3.37
GH06	−15.59	4.87	−14.94	4.69	−16.11	4.42
GH07	−7.11	3.89	−6.74	3.74	−7.62	3.69
GH19	−27.40	4.46	−26.00	4.35	−31.75	3.35
GX12	−20.94	3.33	−19.97	3.15	−22.34	4.03

$v_{i}$: velocity, $s_{i}$: standard deviation of velocity, i: dataset.

**Table 7 sensors-24-00043-t007:** The test and critical values of the significance test. The bold values indicate significant velocities.

Station	$T_{ALOS-2}$	$T_{Sentinel-1\;Asc.}$	$T_{Sentinel-1\;Desc.}$	Critical Value
GH04	0.99	0.98	0.16	3.75
GH05	0.76	0.08	0.57	3.36
GH06	2.07	2.18	3.32	3.36
GH07	0.17	0.18	1.00	3.36
GH19	**6.49**	**5.82**	**9.24**	4.54
GX12	0.54	0.92	0.56	4.54

**Table 8 sensors-24-00043-t008:** Vertical displacement rates of the stations (mm/yr).

Station	$v_{InSAR}$	$v_{PPP}$
GH04	−13.71	−11.70
GH05	−12.21	−14.46
GH06	−4.86	−19.75
GH07	−7.86	−9.50
GH19	−0.27	−37.59
GX12	−23.58	−27.77

## Data Availability

The products presented in this article are available on request from the corresponding author.

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
