# Peer review of "A Statistical Approach for the Integration of Multi-Temporal InSAR and GNSS-PPP Ground Deformation Measurements"

_sensors, 2023, doi:10.3390/s24010043_

Round 1

Reviewer 1 Report

Comments and Suggestions for Authors

The paper is well structured and has studied an interesting target area using both PSI (Sentinel-1 and ALOS) and campaign-mode GNSS measurements. I have made some major and minor comments which I would suggest authors read them carefully for possible revision:

Major comments:

-Where is the reference point (zero velocity) for PSI analysis? InSAR is relative and the displacements of PS points can be different if the location of the reference point is changed (depends on deformation zones).  The chosen reference point should be shown on the proper figure. 

Then, the question is how  the movements of PS points relative to the ref. PS can be compared to GNSS-PPP displacements where all the motions are absolute? This needs a discussion and clarification.

-What was the chosen reference frame or coordinate system for RTX PPP post processing? This information is missing in Section 2.4, however, later in section 3, authors mention the ground displacements are in topocentric coordinates. This should be clear which coordinate system first was used for RTX data processing and how they convert it (or give a reference) to the topocentric coordinates.

-Figure 7: There are only 14 stations that have been measured using GNSS. Not enough dense data to make a grid and interpolate the GNSS velocities for the whole area. Or maybe all colorful zones are interpolated from PSI results? At least the caption should be modified to know what is the background colorful image coming from. The same for Figure 6.

-As noted by authors in line 323, the LOS displacements can be different for ascending and descending results and to make it best understandable, a combination of both ascending and descending datasets, at least for the proper radius (200 m) case, is suggested. This helps to report the combined rates in EW and UD directions which is usually requested by non-InSAR experts. 

-How accurate is RTX-PPP positioning? Perhaps cm-level for 6 hours GNSS data, because it can not model the atmospheric errors relative to 24 hours data. Please discuss it.

Minor comments:

Add period for InSAR data (both for ALOS2 and Sentinel-1) and for campaign-mode GNSS measurements  (2017-2020) in the Abstract. 

Give some numbers for estimated subsidence rate range in the Abstract

Citation in the text: LIne 56, I think, better to start the sentence with the name of authors instead of ref (3). The same for line 80. And good to check line 60, “in the study (4)”. Good to check the instructions for citing previous works in the main text.  

Line 77: check “with Sentinel-1 COSMO-SkyMed data.” did you use COSMo data?

Line 96: “and measurements were not made.” which measurements?

Figure 1, not good quality images, not easy to read the color scales. 

Figure 3, not good quality images, not clear which ones are for S1 and which ones for ALOS. The white text under each image is not readable (I tried to zoom it 172%, still not visible). The best is to have the images readable and clear in 100% zoom.

Figure 4 , not good quality-

Figure 5 caption: “Vertical displacement map” of what? Combined, or obtained from S1 or Alos or GNSS?

Figure 6 caption, the same comment as for Figure 5. The captions should well describe the figures.

Table1: add number of used images for both ascending and descending orbits in the Table.

Table 7: Instead of numbers, it is better to show them in a graph,  for example with vertical bars, to better compare the datasets and radius and stations. 

Good luck

Faramarz Nilfouroushan

Author Response

We would like to thank Reviewer 1 for reviewing our paper. Please find our point by point response below. The responses highlighted with yellow.

General comments

Q1) The paper is well structured and has studied an interesting target area using both PSI (Sentinel-1 and ALOS) and campaign-mode GNSS measurements. I have made some major and minor comments which I would suggest authors read them carefully for possible revision.

A1) Thanks again for your comments. We carefully revised the paper according to your following comments.

Major Comments

Q2) Where is the reference point (zero velocity) for PSI analysis? InSAR is relative and the displacements of PS points can be different if the location of the reference point is changed (depends on deformation zones). The chosen reference point should be shown on the proper figure.

A2) We replotted Figure 3. Reference points for PSI analysis have been shown.

Q3) Then, the question is how the movements of PS points relative to the ref. PS can be compared to GNSS-PPP displacements where all the motions are absolute? This needs a discussion and clarification.

A3) As you mentioned, displacements of the PS points are determined w.r.t reference point in PSI analysis. In the velocity estimation step, we first calculated the Line of Sight (LOS) differences by subtracting the LOS value in first epoch from each LOS values. We also followed the same strategy for GNSS solutions. We begin by converting 3D Cartesian coordinates to topocentric coordinates. Subsequently, we calculate a LOS based on the topocentric coordinates, considering the north, east, and up coordinates. Following this step, we proceed to compare the LOS obtained from InSAR with that derived from GNSS. So, in each technique the differences were used to estimate velocities. In the deformation analysis, the reference points/fiducial stations should be selected from outside the deformation area. Reference points for PSI analysis were also selected from outside the deformation area. PPP solutions are obtained in global reference frame. Hence, it is free of datum inconsistencies since there is no need a network structure. Therefore, PPP has an advantage for determining the crustal and local deformations. For example, AbouAly et al. (2021) used CSRS-PPP which is an online PPP software, to show the usability of PPP for determining the crustal velocity. According to authors, PPP approach can be applied for the investigation of crustal deformation. Yalvac (2020) stated that PPP has become a popular technique showed fast and reliable results for InSAR verification. Hence, the technique can be used for the comparison of GNSS and InSAR.

AbouAly, N., Elhussien, M., Rabah, M. et al. Assessment of NRCAN PPP online service in determination of crustal velocity: case study Northern Egypt GNSS Network. Arab J Geosci 14, 188 (2021). https://doi.org/10.1007/s12517-021-06530-8

Yalvac, S. Validating InSAR-SBAS results by means of different GNSS analysis techniques in medium- and high-grade deformation areas. Environ Monit Assess 192, 120 (2020). https://doi.org/10.1007/s10661-019-8009-8

Q4) What was the chosen reference frame or coordinate system for RTX PPP post processing? This information is missing in Section 2.4, however, later in section 3, authors mention the ground displacements are in topocentric coordinates. This should be clear which coordinate system first was used for RTX data processing and how they convert it (or give a reference) to the topocentric coordinates.

A4) In the service, reference frame can be selected by the user. We selected, ITRF2014. The coordinates and covariance matrices of the results are in a 3D X,Y,Z Cartesian Coordinate System. For the sake of easily interpretation, we transformed the coordinates to a topocentric coordinate system. In this step the coordinates of the first epoch were taken as reference for each station. We added additional below sentences to the text to clarify.

“In the service, coordinate system can be selected by the user. We selected ITRF2014 datum. The coordinates and their covariance matrix are obtained in a 3D Cartesian coordinate system. We then transformed the coordinates and the covariance matrices for each campaign to a topocentric coordinate system w.r.t each station’s first epoch.”

Q5) Figure 7: There are only 14 stations that have been measured using GNSS. Not enough dense data to make a grid and interpolate the GNSS velocities for the whole area. Or maybe all colorful zones are interpolated from PSI results? At least the caption should be modified to know what is the background colorful image coming from. The same for Figure 6.

A5) The horizontal and vertical maps were derived from PSI results. We modified the figure captions.

Q6) As noted by authors in line 323, the LOS displacements can be different for ascending and descending results and to make it best understandable, a combination of both ascending and descending datasets, at least for the proper radius (200 m) case, is suggested. This helps to report the combined rates in EW and UD directions which is usually requested by non-InSAR experts.

A6) Actually, we combined three InSAR datasets to obtain horizontal and vertical deformation for 200 m radius circle. The results were presented at Table 8 (table numbers are updated). Since it was probably not clear enough, we added below sentences to the text.

“One should note that, it is possible to compare the combination/integration of the InSAR analysis with the GNSS. We also integrated the three InSAR datasets in this study. The results of integration were presented in the next section.”

Q7) How accurate is RTX-PPP positioning? Perhaps cm-level for 6 hours GNSS data, because it can not model the atmospheric errors relative to 24 hours data. Please discuss it.

A7) Thanks for the pointing this subject. In the literature, there are many studies investigate the accuracy of GNSS. Sezer et al. (2021) formulated the error of topocentric coordinates following the strategies used in Saracoglu and Sanli (2020). In the study, accuracy functions were determined for Multi constellation systems. For the GPS and GLONASS systems, accuracies can be determined as;

North = 6.38 / sqrt(T)

East = 8.97 / sqrt(T)

Up = 18.70 / sqrt(T)

T is session duration in hour. Hence with 6 hour observations, the accuracies are 2.60, 3.66 and 7.63 mm for north, east and up, respectively. For all components, accuracies are mm level. Since the deformations are cm level, 6-hour observation can seem as acceptable.

We added additional below sentences to the text to clarify.

“Moreover, a 6-hour observation session duration, as indicated by Sezer et al. [20], employ-ing the accuracy function provided in Saracoglu and Sanli [21], can yield millimeter-level accuracy.”

 [21] Saracoglu, A. and Sanli, D. U. (2020). Effect of meteorological seasons on the accuracy of GPS positioning, Measurement 152, 107301.

Minor Comments

Q8) Add period for InSAR data (both for ALOS2 and Sentinel-1) and for campaign-mode GNSS measurements (2017-2020) in the Abstract.

A8) The periods were added.

Q9) Give some numbers for estimated subsidence rate range in the Abstract

A9) The sentence given below was added in the abstract.

“For the coherent stations, vertical displacement rates vary between -4.86 mm/yr and -23.58 mm/yr and, -9.50 and -27.77 mm/yr, for InSAR and GNSS, respectively.”

Q10) Citation in the text: LIne 56, I think, better to start the sentence with the name of authors instead of ref (3). The same for line 80. And good to check line 60, “in the study (4)”. Good to check the instructions for citing previous works in the main text. 

A10) The text was corrected.

Q11) Line 77: check “with Sentinel-1 COSMO-SkyMed data.” did you use COSMo data?

A11) In this study, COSMO data was not utilized. However, when referencing the study conducted by Fabris et al., we mentioned COSMO SkyMed data, as they employed it in their research.

Q12) Line 96: “and measurements were not made.” which measurements?

A12) The sentence was corrected.

Q13) Figure 1, not good quality images, not easy to read the color scales.

A13) The figure was replotted.

Q14) Figure 3, not good quality images, not clear which ones are for S1 and which ones for ALOS. The white text under each image is not readable (I tried to zoom it 172%, still not visible). The best is to have the images readable and clear in 100% zoom.

A14) The figure was replotted.

Q15) Figure 4, not good quality-

A15) The figure was replotted.

Q16) Figure 5 caption : “Vertical displacement map” of what? Combined, or obtained from S1 or Alos or GNSS?

A16) The caption related to Figure 5 (now in the revised version Figure 6) was modified.

Q17) Figure 6 caption, the same comment as for Figure 5. The captions should well describe the figures.

A17) The caption was modified.

Q18) Table1: add number of used images for both ascending and descending orbits in the Table.

A18) We added the number of images in the Table.

Q19) Table 7: Instead of numbers, it is better to show them in a graph, for example with vertical bars, to better compare the datasets and radius and stations.

A19) In Figure 4 we added bar graph of correlation values instead of Table 7.

Reviewer 2 Report

Comments and Suggestions for Authors

The authors utilized InSAR and GNSS technology to monitor land subsidence along the Golden Horn coast. While the manuscript shows promise, there are several issues that need to be addressed before it can be considered for publication.

1) The resolution of the figures in the manuscript is insufficient, leading to some information being indiscernible. The authors should ensure that images are not compressed when editing documents.

2) In Line 126, two types of SAR data, ALOS-2 and Sentinel-1, are employed. The coverage of the images should be indicated in a figure.

3) It is recommended to use the term "primary image" instead of "master image."

4) Figure 2 displays only one picture, and the last word (right) seems redundant.

5) In Lines 225-230, please rewrite the sentences in a more professional manner.

6) Lines 249-251: Why filter GNSS data before comparison instead of using all data?

7) Lines 269-270: It should be clarified the correlations are between which datasets.

8) In Line 310 and Table 7, generally speaking, it is uncommon for the correlation to exceed 0.98. Please reevaluate the data. Additionally, provide the statistical sample size. High correlations often occur when the number of samples is small. This situation lacks statistical significance.

9) In Table 9, the difference between the two results on the GH19 station is substantial. This leads to divergent conclusions: -0.27mm/yr suggests stability, while -37.59mm/yr indicates significant deformation. This discrepancy should be addressed to ensure consistent and reliable findings.

Comments on the Quality of English Language

Moderate editing of English language required.

Author Response

We would like to thank Reviewer 2 for reviewing our paper. Please find our point by point response below. The responses highlighted with green.

General comments

  1. R) The authors utilized InSAR and GNSS technology to monitor land subsidence along the Golden Horn coast. While the manuscript shows promise, there are several issues that need to be addressed before it can be considered for publication.
  2. A) Thanks again for your comments. We carefully revised the paper according to your following comments.

Detailed Comments

Q1) The resolution of the figures in the manuscript is insufficient, leading to some information being indiscernible. The authors should ensure that images are not compressed when editing documents.

A1) The figures were replotted.

Q2) In Line 126, two types of SAR data, ALOS-2 and Sentinel-1, are employed. The coverage of the images should be indicated in a figure.

A2) Fig. 1 was replotted.

Q3) It is recommended to use the term "primary image" instead of "master image."

A3) We edited the text regarding to your comment.

Q4) Figure 2 displays only one picture, and the last word (right) seems redundant.

A4) The word was deleted.

Q5) In Lines 225-230, please rewrite the sentences in a more professional manner.

A5) We edited the text as below.

“Calò et al. [4] employed one and a half years of TerraSAR-X data spanning from November 2010 to June 2012, utilizing the Small Baseline Subset (SBAS) method. Their analysis revealed deformation in the coastal areas of the GH, resulting in a displacement of approximately 4-5 cm (an average of ~3 cm/year). Moreover, Imamoglu et al. [2] reported a displacement rate of 8 mm/year based on Sentinel-1 data collected be-tween 2014 and 2017. Additionally, Aslan et al. [1] determined a maximum displacement of 10 mm/year using Sentinel-1 data for the same period. The present study corroborates these findings, indicating displacement values in the same region, particularly within the latter section of the GH. Furthermore, it is observed that the observed movement persisted in the region after 2017.”

Q6) Lines 249-251: Why filter GNSS data before comparison instead of using all data?

A6) Actually, the simple answer is to not cause the wrong interpretation of the results. As you know, all kind of measurements include errors. These errors can be random errors, can be neglected, and outlier. Since the GNSS observations also include errors, the results of the analysis can be erroneous. As stated answer of the above question and in the beginning of the “Results and Discussion” section, we expect subsidence in the study region. Hence, we first defined the stations which have significant vertical displacement rate to avoid wrong interpretation and wrong comparison.

Q7) Lines 269-270: It should be clarified the correlations are between which datasets.

A7) The correlation values are between InSAR datasets and GNSS as stated previous sentence. According to your comment, we edited the sentences as below.

“Correlations between ALOS-2 and GNSS data are up to 0.72, 0.84, and 0.78 values, respectively. Correlations between Sentinel Descending and GNSS are 0.65, 0.75, and 0.61; between Sentinel-1 Ascending and GNSS, they are 0.65, 0.82 and 0.75.”

Q8) In Line 310 and Table 7, generally speaking, it is uncommon for the correlation to exceed 0.98. Please reevaluate the data. Additionally, provide the statistical sample size. High correlations often occur when the number of samples is small. This situation lacks statistical significance.

A8) Thanks for pointing this issue. Actually, you are right. The number samples affect the correlation value. In the study region, there were no stations belong to the any continuously operating reference station network. Therefore, we designed a new network. And campaign type of the observations were obtained for the stations. To determine the velocity, at least three campaigns is required. For the acceleration, at least four campaigns are required. We performed seven campaigns. And, we did not use the observations of the stations which have less than observations from 3 campaigns. We preformed seven campaigns for this study. Therefore, the results of the study can be seem as acceptable. Moreover, Table 7 was removed, and correlation values are shown with a bar graph (Figure 4) according to Reviewer 1’s comment.

Q9) In Table 9, the difference between the two results on the GH19 station is substantial. This leads to divergent conclusions: -0.27mm/yr suggests stability, while -37.59mm/yr indicates significant deformation. This discrepancy should be addressed to ensure consistent and reliable findings.

A9) Thanks for pointing this issue. As you mention, the difference between the techniques is too high. After the third campaign, a building activity was started too close to the GH19 station. The high difference can be related to this. Also, the GNSS technique can determine only the station’s movement not the region’s. The movement of the station does not mean that there is also a movement in a point 20 meter close to the station. Moreover, we used PSI analysis for processing of the InSAR datasets. In PSI analysis, deformations are determined based on the PS points’ displacements. Therefore, if any PS point does not overlap on the GNSS station, the movement can not be determined. So, it might be related with the local effects. 

Round 2

Reviewer 1 Report

Comments and Suggestions for Authors

Thank you for addressing my comments. Still, I am not happy with the figures and captions, figure captions should have enough information to explain the figures. Please see below (and also see my comment for ALOS2):

1- Figure 1, what are those rectangles? (I know what they are, but this info is missing in the caption).

2- Figure 4, what are tjsoe strange numbers 150 m, 200m, 250 m in the legend of the figures? explanation is missing.

3-Figure 6, this is interpolated displacement map generated from interpolation of PS displacements. The same for Figure 7 and 8, "interpolated" is missing....

- And why only Ascending dataset for ALOS2 was used? and any challenge for combiantion of the data with S1 which has both ascending and descending datasets? 

Author Response

We would like to thank Reviewer #1 for reviewing our paper.

Please find our point by point response below.

General comments and suggestions

R) Thank you for addressing my comments. Still, I am not happy with the figures and captions, figure captions should have enough information to explain the figures.

A) Thanks for your comments. We edited the figure captions. Also, we replotted the Figures 5 and 6 according to your comments. Also, we removed the Figure 8 to avoid repetition.

R) 1- Figure 1, what are those rectangles? (I know what they are, but this info is missing in the caption).

A) Figure caption was edited.

R) 2- Figure 4, what are tjsoe strange numbers 150 m, 200m, 250 m in the legend of the figures? explanation is missing.

A) The explanation was added.

R) 3-Figure 6, this is interpolated displacement map generated from interpolation of PS displacements. The same for Figure 7 and 8, "interpolated" is missing....

A) Figure caption was edited.

R) And why only Ascending dataset for ALOS2 was used? and any challenge for combiantion of the data with S1 which has both ascending and descending datasets?

A) The number of ALOS-2 images provided to us by JAXA for the descending track was not sufficient to obtain reliable LOS-projected ground displacements. Also, the simple strategy adopted relies on the solution of a determined system of three linear equations in the 3-D (Up-Down, east-west, north-south) ground displacement rates unknowns. Accordingly, it works well with three independent and complementary datasets. Note that in our work we followed the simple but effective strategy in [31] based on the assumption that the deformation is almost linear. The use of ALOS-2 was to better constrain the solution, considering the known sensitivities of the SAR system to the 3-D ground displacement rates. A few additional clarifications have coincisely been reported in Section 3.2.  

Reviewer 2 Report

Comments and Suggestions for Authors

The authors have revised the manuscript according to my comments. It is ready for publication.

Comments on the Quality of English Language

Minor editing of English language required

Author Response

Thanks